# Hair Thickness Growth Effect of Adenosine Complex in Male-/Female-Patterned Hair Loss via Inhibition of Androgen Receptor Signaling

**DOI:** 10.3390/ijms25126534

**Published:** 2024-06-13

**Authors:** Jaeyoon Kim, Jae young Shin, Yun-Ho Choi, Jang Ho Joo, Mi Hee Kwack, Young Kwan Sung, Nae Gyu Kang

**Affiliations:** 1LG Household & Health Care (LG H&H) R&D Center, 70, Magokjoongang 10-ro, Gangseo-gu, Seoul 07795, Republic of Korea; kjy5281@lghnh.com (J.K.); sjy2811@lghnh.com (J.y.S.); youknow@lghnh.com (Y.-H.C.); janghojoo@lghnh.com (J.H.J.); 2Department of Immunology, School of Medicine, Kyungpook National University, Daegu 41944, Republic of Korea; go3004@knu.ac.kr (M.H.K.); ysung@knu.ac.kr (Y.K.S.)

**Keywords:** adenosine, anti-androgenic activity, hair thickness, androgen receptor, androgen alopecia

## Abstract

Aging (senescence) is an unavoidable biological process that results in visible manifestations in all cutaneous tissues, including scalp skin and hair follicles. Previously, we evaluated the molecular function of adenosine in promoting alopecia treatment in vitro. To elucidate the differences in the molecular mechanisms between minoxidil (MNX) and adenosine, gene expression changes in dermal papilla cells were examined. The androgen receptor (AR) pathway was identified as a candidate target of adenosine for hair growth, and the anti-androgenic activity of adenosine was examined in vitro. In addition, ex vivo examination of human hair follicle organ cultures revealed that adenosine potently elongated the anagen stage. According to the severity of alopecia, the ratio of the two peaks (terminal hair area/vellus hair area) decreased continuously. We further investigated the adenosine hair growth promoting effect in vivo to examine the hair thickness growth effects of topical 5% MNX and the adenosine complex (0.75% adenosine, 1% penthenol, and 2% niacinamide; APN) in vivo. After 4 months of administration, both the MNX and APN group showed significant increases in hair density (MNX + 5.01% (*p* < 0.01), APN + 6.20% (*p* < 0.001)) and thickness (MNX + 5.14% (*p* < 0.001), APN + 10.32% (*p* < 0.001)). The inhibition of AR signaling via adenosine could have contributed to hair thickness growth. We suggest that the anti-androgenic effect of adenosine, along with the evaluation of hair thickness distribution, could help us to understand hair physiology and to investigate new approaches for drug development.

## 1. Introduction

Aging (senescence) is an unavoidable biological process with visible manifestations in all cutaneous tissues, including hair follicles. Hair aging comprises both intrinsic aging, which includes natural physiological changes in hair follicles and surrounding skin tissues, and extrinsic aging, which includes changes associated with environmental exposure and physical stress. The most well-recognized signs of intrinsic aging are hair loss (alopecia) and graying. However, hair aging changes go far beyond hair number and color alterations, along with many other hair properties, such as the diameter, shape, growth patterns, and mechanical and tactile aspects [1,2].

Alopecia could be classified by its causes, such as disease, mechanical stress, nutritional deficiency, hormone imbalance, and aging [3,4]. Although diverse treatment strategies have been suggested for hair loss, their efficacies remain limited. Minoxidil (MNX) and finasteride have been approved by the US Food and Drug Administration and are widely used for alopecia treatment [5]. Currently, Baricitinib, a reversible inhibitor of Janus kinase 1/2, has been approved for treating alopecia areata [6]. MNX, the most widely administered drug initially developed for hypertension treatment, was discovered to promote hair density. MNX has been identified as a potassium channel opener; however, its exact hair growth mechanism is not fully understood [7].

Adenosine, a ubiquitous purine nucleoside produced via adenosine monophosphate (AMP) dephosphorylation, plays an essential role in tissue protection and recovery [8,9]. Extracellular adenosine directly activates four adenosine receptor subtypes: A_1_, A_2A_, A_2B_, and A_3_ [10]. The adenosine A2A and A2B receptor signaling pathways have been suggested as potential targets for alopecia treatment [11], and one of the mechanisms of MNX for improving alopecia is regulating the adenosine receptor signaling via SUR2B receptor activation in dermal papilla cells (DPCs) [12]. We have previously reported that adenosine activated the Wnt/β-catenin pathway in human dermal fibroblasts through the activation of MAP kinases such as MEK1/2, mTOR, and p70S6K, and the inhibitory phosphorylation of Gsk3β Ser9 site was suggested as a key mechanism for the adenosine-mediated Wnt/β-catenin pathway activation [13]. The Gαs/cyclic AMP/PKA/mTOR cascade plays a pivotal role in adenosine-mediated Wnt pathway activation, being the underlying molecular mechanism of adenosine for its hair growth-promoting effect [14].

The hair follicle is one of the most energy-consuming organs in humans. Therefore, the energy demand is high, particularly during the anagen stage [15]. In this context, maintaining high rates of energy metabolism may be beneficial for preventing alopecia. For example, enhanced mitochondrial potential in ex vivo experiments with human hair follicles (hHF) stimulated hair follicle growth and prevented the anagen–catagen transition [16,17,18]. Because mitochondrial energy generation in DPCs accounts for the majority of the energy metabolism in cells [15], the induction of mitochondrial activity might drive human hair growth both in vitro and in vivo [19,20].

In previous research, we have found that adenosine treatment in cultured human DPCs (hDPCs) increased NADH generation and mitochondrial membrane potential (∆Ψ), which are strongly associated with the positive steady state of NAD/NADH coupling [14]. In addition, we investigated hair growth reinforcement via three vitamins of the vitamin B complex: D-panthenol, which stimulates energy metabolism in hDPCs [21]; niacinamide, which protects hDPCs from oxidative stress [22]; and riboflavin 5 phosphate, which exhibits anti-androgenic activity by downregulating androgen receptors (ARs) [23].

In patients with alopecia, both hair density and diameter decrease with alopecia severity [24]. The effect of MNX, a widely used US Food and Drug Administration-approved drug, on hair density enhancement was observed, but reports on its effect on hair thickness are limited [25]. Furthermore, even though the technology for measuring hair thickness has been continuously developed [26,27], guidelines for measuring hair thickness in alopecia to reflect scalp physiological properties are limited.

In this study, we investigated the molecular mechanisms underlying the anti-androgenic activity of adenosine in cultured hDPCs and evaluated the effects of topically administered MNX and the adenosine complex (adenosine, panthenol, and niacinamide (APN)) on hair density and thickness.

The basal expression levels of adenosine receptor subtypes A_1_, A_2A_, A_2B_, and A_3_ were examined, and gene expression in response to adenosine treatment was evaluated in cultured hDCPs. The AR was significantly correlated with adenosine, and the anti-androgenic activity of adenosine was examined. In addition, adenosine elongated the hair anagen stage in cultured human hair follicles. Topical administration of MNX and APN for four months was also investigated, and hair density and thickness were evaluated. Our data strongly suggest that APN possesses therapeutic potential for preventing and/or treating hair loss. Changes in hair thickness distribution via MNX and APN could help to understand hair physiology and investigate new approaches for drug development.

## 2. Results

### 2.1. Adenosine Receptor A_2B_ Is a Dominant Subtype in Cultured Human Dermal Papilla 

In a previous study, we examined four G protein-coupled adenosine receptor subtypes, A_1_, A_2A_, A_2B_, and A_3_, in human dermal fibroblast cells [13]. The selectivity of adenosine and NECA for each adenosine receptor has been reported previously [10,28]. Before investigating the effects of adenosine on hDPCs, we analyzed the mRNA expression levels of adenosine receptor subtypes, ADORA1, ADORA2A, ADORA2B, and ADORA3, in cultured hDPCs. The adenosine receptor A_2B_ showed the highest expression level among the subtypes, approximately 2.77-fold higher than that of ADORA2A. In contrast, mRNA expression of the A_3_ subtype was not detected (Figure 1b). Therefore, it could be presumed that the physiological responses induced with adenosine in cultured hDPCs were mediated by the A_2A_ and A_2B_ adenosine receptor subtypes. 

### 2.2. Activation of Signaling Pathways via MNX and Adenosine 

To understand the molecular mechanism of action of adenosine, we investigated the changes in mRNA expression in hDPC treated with MNX and adenosine. We examined 78 genes in six categories: cell differentiation, cell junction, cellular survival, cytoskeleton, growth factor signaling, and tissue development (Appendix A). Among the 78 genes, several were significantly upregulated or downregulated. For example, decorin, KI67, VEGFA, and Bcl2 were increased with MNX and adenosine, but DKK1, TGFb1, and BMP4 were decreased (Figure 2a). A total of 5 genes were significantly altered by MNX, and 11 genes were altered by adenosine alone. Sixteen genes showed more than a 1.5-fold increase with statistical significance (*p* < 0.05). They were selected for further investigation, namely, protein clustering analysis [29] and pathway estimation following Gene Ontology and Kyoto Encyclopedia of Genes and Genomes analyses (Figure 2b,c, Appendix A).

MNX- and adenosine-activated pathways related to cellular metabolism and proliferation (Figure 2b, Appendix A). Among these, the PI3K-AKT and Ras signaling pathways were predicted to be regulated by MNX and adenosine. In addition, adenosine stimulated organ development processes, such as the Hippo and Wnt signaling pathways, but MNX did not (Figure 2c). Furthermore, based on in silico modeling, interacting proteins such as AR, E-/N-cadherin, and presenilin-1 were predicted (Table 1). These estimations could help understand the molecular mechanisms underlying hair physiology for thickness growth.

### 2.3. Anti-Androgenic Effect of Adenosine 

Following in silico modeling analysis, AR, but not MNX, was estimated to be one of the proteins highly correlated with adenosine treatment. Finasteride, one of the drugs for treating androgenic alopecia, inhibits AR activation by interfering with 5-alpha reductase activity [5,30]. Therefore, we investigated the anti-androgenic activity of adenosine in 22Rv1-F5-MMTV/Luc cells. Adenosine showed anti-androgenic activity from a concentration of 30 μM (Figure 3a).

In addition, downstream of the AR signaling pathway, the target gene expression of SRD5A1 and SRD5A2, coding for 5-alpha reductase type 1 and 2, respectively, were examined in hDPCs. SRD5A2 expression levels were significantly decreased with adenosine treatment (Figure 3b). Furthermore, the phosphorylation of kinase proteins (p53, Hsp27, JNK, and MKK3/6), which contribute to the AR pathway [31], was evaluated (Figure 3c). Phosphorylation of p53(S16), MKK3(S189), and MKK6(S207) was significantly increased with adenosine treatment, whereas that of Hsp27(S82) and JNK(T183) was downregulated (Figure 3d). The downregulation of AR activity by adenosine can contribute to hair growth. 

### 2.4. Adenosine Elongated Anagen Stages in an Ex Vivo Hair Follicle Organ Culture

The effect of adenosine on the hair follicle stage was investigated in a human hair follicle (hHF) organ culture model. At the end of the 6-day incubation period, 11.1% of hHFs remained in the anagen stage in the non-treated control group. In contrast, 94.4% of the hHFs treated with adenosine (0.45%) remained in the anagen stage (Figure 4a), comparable to the MNX-treated control group. Adenosine caused cultured hHFs to retain the characteristic morphology of the anagen stage; therefore, the proportion of hHFs in the anagen stage tended to increase with adenosine concentration. In addition, we evaluated KI67 levels in hHF treated with adenosine. The KI67 level in the adenosine treatment group increased, especially around the dermal papilla (Figure 4b). The positive control, MNX, showed comparable results.

### 2.5. Hair Thickness Analysis following Alopecia Scale and Senescence

To evaluate the properties of the scalp and hair physiology, we investigated hair density and thickness in 156 volunteers with diverse alopecia severities (Appendix A). Hair shaft images (average 84.9 shafts in a 1 cm^2^ area) were randomly sampled from the crown area of each participant (Appendix A). Hair density and thickness decreased with the severity of alopecia (Figure 5). However, in the Korean cohort, the patterns of decreased hair density and thickness were different. Hair density significantly decreased from HN I to HN II, but there was a limited change from HN II to HN V–VI (Figure 5b). In contrast, hair thickness continuously decreased from HN I to V–VI, and, especially, the thickness distribution between the HN III and HN IV groups changed significantly (Figure 5a,c). 

The physiological characteristics of the scalp of Korean participants aged 20s–60s were evaluated. The physiological parameters changed with age (Figure 5d,e). Hair thickness was significantly correlated with senescence (*p* = 0.022). Hair density and scalp skin barrier (transepidermal water loss (TEWL)) continuously decreased with age. Hair thickness and scalp skin elasticity decreased significantly from 20s to 30s (Table 2). These changes in scalp physiological characteristics, hair density, thickness, scalp skin elasticity, and barrier reflect scalp senescence due to aging and could help to understand hair physiology.

The randomly sampled hair shaft thickness distribution of each participant changed according to the Hamilton–Norwood (HN) alopecia scale. Hair shaft thickness distribution showed two peaks (one representing terminal hair and the other representing vellus hair). According to the severity of alopecia, the ratio of the two peaks (terminal hair area/vellus hair area) decreased continuously (Figure 5a).

### 2.6. Hair Thickness Enhancement with MNX and APN Administration In Vivo

To evaluate the effect of MNX and APN on hair thickness, their topical administration in a Korean alopecia cohort was investigated. No significant differences in baseline characteristics, including age (*p* = 0.8288), baldness grade (*p* = 0.5979), hair density (*p* = 0.8124), and hair thickness (*p* = 0.3423), were observed between the two groups (Appendix A).

After four months of administration, hair density and thickness increased in both the MNX and APN groups. Hair density continuously increased in the MNX and APN groups over four months (Figure 6a,b, and Table 3). However, the distribution of changes in hair shaft thickness also differed between MNX and APN groups. For MNX, the thickness of the overall range shaft increased. In contrast, in the APN group, the proportion of vellus hair decreased, and that of terminal hair increased (Figure 6d). Therefore, we propose that the mechanisms by which MNX and adenosine contribute to hair thickness differ. The anti-androgenic effects of adenosine may have contributed to hair thickness growth.

## 3. Discussion

In this study, we investigated the effects of topical administration of MNX and APN on the physiological parameters of the scalp, including senescence and hair thickness. Scalp skin health, such as hair density, thickness, scalp skin barrier, and elasticity, declines with age, similar to facial skin [32]. As described previously, the pattern of decrease in each scalp parameter varied with age (Table 2). In addition, pattern changes in scalp parameters were observed to be different between male and female participants. 

Hair shaft evaluation methods were examined to understand the physiology of hair thickness. In this study, we photo-documented hair shaft images with random sampling in a 1 cm^2^ area of the scalp crown region. When hair thickness was evaluated using 5–40 samples, there was a possibility of a significant difference in the measured values (Appendix A). Therefore, in this study, we sampled more than 50 hair shaft images from each participant and evaluated an average of 84.8 hair shaft images for hair thickness. 

Following hair thickness evaluation, shaft growth characteristics were observed. As shown in Figure 5, the hair thickness patterns differed according to the severity of alopecia. Average values of thickness were drastically decreased from HN I to II (from 71.3 to 59.7 μm). In contrast, the hair thickness patterns changed significantly from HN III to HN IV. The ratio between the vellus hair and terminal hair in the alopecia group changed. The vellus/terminal hair ratio in androgen alopecia was increased in previous reports [33,34]. Lowering the vellus/terminal hair ratio is a candidate strategy for alopecia treatment.

Furthermore, the investigation of hair thickness could help in understanding hair follicle physiology in more detail. The decrease in hair density was a critical issue in the early stage of alopecia, and a decrease in hair thickness was essential in the late stage of alopecia in the Korean cohort (Figure 5). Therefore, hair thickness distribution and density changed according to the severity of alopecia. Consequently, scalp care for hair thickness and density is required according to the severity of alopecia. 

Using a hair thickness measurement method, we investigated the effects of MNX and APN on hair density and thickness. In this study, after four months of administration, hair density significantly increased in the MNX (from 168.1 to 176.5 number/cm^2^) and APN (from 168.1 to 176.5 number/cm^2^) groups (Table 3). In previous reports, the non-vellus hair density increased by 24.5 number/cm^2^ from 151.1 (±45.9) number/cm^2^ after 48-week (12 months) administration of 5% MNX [7]. On the contrary, in Ramos et al., 2019, 5% topical MNX promoted increased total hair density from 163.2 (±46.0) to 176.3 (±61.5) number/cm^2^, but the density increase for the terminal hair was limited (from 113.3 (±41.1) to 116.8 (±44.9) number/cm^2)^ for six-month administration [35]. The recovery rate of hair density is relatively limited during half of the year. However, hair thickness significantly increased after 2-month administration. It is possible that the mode of the mechanism differs between hair density and thickness. In this study, the investigation period (four months) was limited for comparison with other reports (six months and one year). Further long-term research (one year or more) on MNX and APN could help understand scalp physiology related to hair thickness growth. 

As mentioned above, the changing patterns of thickness were different (Figure 6). For the MNX group, the overall hair shaft was thicker. These results suggest that MNX has a growth-promoting effect on hair follicles and contributes to hair shaft growth during the anagen period [25]. MNX stimulated cell proliferation and metabolism. Furthermore, MNX activated the Ras and PI3K-AKT signaling pathways (Figure 2). Ras and MAPK signaling are correlated with hair follicular cell proliferation and may contribute to hair growth [36]. Therefore, MNX may affect hair thickness growth rate via the PI3K-AKT signaling pathway.

In contrast, the portion of vellus hair decreased, and the average hair thickness was enhanced with APN treatment (Figure 6). Adenosine is a purinergic nucleoside that is involved in various cellular and physiological processes. The topical administration of adenosine is under clinical investigation for wound-healing deficiency and alopecia [10,37]. In the present study, we examined the effects of adenosine on hair density and thickness. In particular, the ratio between the vellus and terminal hair was altered with adenosine treatment (Figure 6). However, the mechanism of the change between the vellus and terminal hair is limited, and its contribution to hair thickness requires further study.

Furthermore, in Figure 3, we examined anti-androgenic activity and the response elements correlated with the AR pathway. SRD5A2, a downstream target gene of the AR [38], was downregulated. p53 negatively regulates the AR responses [39]. The MKK3/6 is negatively correlated with AR activation in prostate cancer cell lines [31,40]. The inactivation of JNK phosphorylation also decreases AR nuclear translocation by inhibiting Sp1 [41,42]. Phosphorylation of Hsp27, a downstream target of the AR, is induced by androgen activation [43]. The increased phosphorylation of p53 and MKK3/6 and decreased phosphorylation of Hsp27 and JNK could be correlated with the downregulation of the AR pathway. DKK-1, which shows high levels of expression in bald scalps with androgen alopecia [44], is upregulated by AR pathway activation [22]. As shown in Figure 2, the DKK-1 levels significantly decreased following adenosine treatment. In addition, decorin, which is inhibited by AR activity [45], was significantly upregulated by adenosine (Appendix A). Therefore, because previous reports on AR were based on prostate cancer, further investigation of the AR pathway in hDCPs is required to elucidate the anti-androgenic activity of adenosine.

In previous research, we investigated whether the molecular mechanism of adenosine and adenosine-mediated Wnt/β-Catenin signaling activation could help to elongate the anagen period [14]. Adenosine stimulates the expression of fibroblast growth factors 2 and 7 (FGF2 and FGF7) in cultured DPCs, which are responsible for hair growth via adenosine A_2B_ activation [46,47]. The regulation of the Wnt/β-catenin pathway is important in the hair follicle cycle and its control [48]. Activation and downregulation of the β-catenin pathway are critical for maintaining the anagen stage and for anagen to catagen transition, respectively [49]. Furthermore, Wnt signaling plays a critical role in the telogen-anagen transition of the human hair cycle by activating quiescent hair follicle stem cells [50]. Therefore, adenosine-mediated signaling activation could contribute to both AR and Wnt/β-catenin signaling, and it will be a candidate for the decrease in the vellus/terminal hair ratio via adenosine. 

In addition, adenosine showed significant anti-androgenic activity (Figure 3). The anti-hair loss effect of adenosine was previously reported to improve androgenetic alopecia and female-pattern hair loss following topical application [37,51,52]. Androgenetic alopecia is a hereditary androgen-dependent progressive thinning of the scalp hair by androgen-dependent processes due to the binding of dihydrotestosterone (DHT) to the AR [53].

Finasteride, a drug for androgenic alopecia, works as a competitive inhibitor of type 2 5-alpha reductase, blocks the conversion of testosterone to DHT, and increases anagen hair [54,55]. As shown in Figure 3, adenosine decreased the expression of type 2 5-alpha reductase in cultured DPCs. Consequently, the anti-androgenic activity of adenosine could be another candidate molecular mechanism to manage vellus hair into terminal hair, and further investigation of finasteride will help to understand the hair growth physiology of shaft thickness.

Diverse vitamins have been suggested for the supplementation of scalp and hair health. One of the most widely used vitamins for hair loss is biotin (vitamin B7). Although biotin has a limited effect on alopecia treatment in clinical trials, biotin deficiency in infants leads to hair loss, and biotin supplementation alleviates alopecia symptoms [56]. Additionally, Vitamin D has been reported to be an important ingredient in maintaining scalp health, and vitamin D deficiency causes hair loss [57]. In a previous study, we investigated the hair growth-promoting effects of panthenol (vitamin B5) and niacinamide (vitamin B3). Panthenol stimulates cell proliferation and reduces the expression of apoptotic and senescent markers, increasing the expression of anagen markers in cultured hDPCs [21]. Niacinamide decreases H_2_O_2_-induced intracellular reactive oxygen species production and DKK-1 expression, which promotes hair follicle regression by inducing catagen [22].

## 4. Materials and Methods

### 4.1. Dermal Papilla Cell Culture 

hDPCs were obtained from PromoCell (Heidelberg, Germany). DPCs were cultured in basal medium supplemented with 4% fetal calf serum, 0.4% bovine pituitary extract, 1 ng/mL basic fibroblast growth factor, and 5 μg/mL insulin (Supplement Mix, PromoCell, Heidelberg, Germany). Cells were maintained in a humidified incubator at 37 °C and 5% CO_2_. Before adenosine (Sigma Aldrich, St. Louis, MO, USA) treatment, serum limitation was achieved by replacing the medium with fresh DMEM (Gibco, Waltham, MA, USA) supplemented with 1% fetal bovine serum (FBS) (Gibco, Waltham, MA, USA) and 1 ng/mL basic FGF (Merck, Darmstadt, Germany) and culturing for 24 h to minimize the effects of serum and growth supplements.

### 4.2. Quantitative Real-Time PCR 

Adenosine was added at concentrations of 1.5 and 3 mM for 24 h, with non-treated cells serving as the control. Total RNA was extracted using an RNeasy RNA extraction kit (Qiagen Inc., Hilden, Germany). cDNA was synthesized using a cDNA synthesis kit (Philkorea, Seoul, Republic of Korea) with a ThermoCycler (R&D Systems, Minneapolis MN, USA) according to the manufacturer’s protocol. cDNA samples obtained from control and treated cells were subjected to real-time (RT) PCR analysis. 

TaqMan probes for RT-PCR used in this study were as follows: GAPDH assay id 4352934E; DCN assay id Hs00754870_s1; KI67 assay id Hs04260396_g1; VEGFA assay id Hs00900055_m1; BCL2 assay id Hs00608023_m1; DKK1 assay id Hs00183740_m1; TGFB1 assay id Hs00998133_m1; BMP4 assay id Hs03676628_s1; AR assay id Hs00171172_m1; SRD5A1 assay id Hs00971645_g1; and SRD5A2 assay id Hs00936406_m1. 

The TaqMan One-Step RT-PCR Master Mix Reagent (Life Technologies, Carlsbad, CA, USA) was used. PCR was performed using an ABI 7500 Real-Time PCR system following the manufacturer’s instructions. The resulting data were analyzed using ABI 7500 software. 

### 4.3. Reporter Assay Using 22Rv1-F5-MMTV/Luc Stable Cell Line

22Rv1 cells were purchased from the American Type Culture Collection (Manassas, VA, USA). The 22Rv1-F5-MMTV/Luciferase (Luc) stable cells were constructed according to previously reported protocols [23]. 

22Rv1, LNCaP, and 22Rv1-F5-MMTV-Luc cells were routinely maintained in RPMI1640 medium (Gibco, Waltham, MA, USA) with 10% FBS (Gibco, MA, USA) and 1% penicillin/streptomycin (Sigma Aldrich, St. Louis, MO, USA) in a humidified atmosphere containing 5% CO_2_ at 37 °C. For the androgen-deprived experimental procedures, cells were cultured in phenol red-free RPMI 1640 (Gibco, Waltham, MA, USA) supplemented with 5% charcoal-stripped FBS (Sigma Aldrich, St. Louis, MO, USA).

Established stable cells were seeded at 2.5 × 10^4^ cells/well in 96-well flat clear-bottom black plates (Sigma Aldrich, St. Louis, MO, USA) and cultured for 48 h. Cells were treated with various concentrations of adenosine in the presence of 1 nM DHT (Sigma Aldrich, St. Louis, MO, USA) for 24 h. For short-term exposure to adenosine, cells were treated for 4 h. After treatment, cells were lysed with 30 μL of 1× passive lysis buffer, and luciferase activity was measured using the luciferase assay system (Promega, Madison, WI, USA).

### 4.4. Protein Dot Blot Analysis for Human MAPK Phosphorylation

A human MAP kinase phosphorylation antibody array kit (Abcam, Cambridge, UK) was used to elucidate changes in signal transduction pathways in DPCs. In total, 17 human MAP kinase phosphorylations were analyzed. Cells were treated with 1.5 and 3 mM adenosine for an appropriate time and collected for MAPK phosphorylation analysis. Cells treated with vehicle medium were used as untreated controls. Conventional immunoblotting was performed according to the manufacturer’s protocol. The resulting blots were analyzed under the same conditions using an iBright FL1000 (Invitrogen, Waltham, MA, USA).

### 4.5. Human Hair Follicle Organ Culture and Hair Cycle Scoring

Human scalp skin was obtained from the non-balding areas of patients undergoing hair transplant surgery with written consent and approval from the Institutional Review Board of Kyungpook National University (IRB No. KNU. 2023-0248). Human hair follicles were isolated via microscopic microdissection. Anagen VI hair follicles were used in the present study. Isolated hair follicles were maintained in William’s E medium (Life Technologies, Carlsbad, CA, USA) supplemented with 10 μg/mL insulin (Sigma Aldrich, St. Louis, MO, USA), 10 ng/mL hydrocortisone (Sigma Aldrich, St. Louis, MO, USA), 2 mM L-glutamine (Life Technologies, CA, USA), 10 U/mL penicillin (Life Technologies, Carlsbad, CA, USA), 100 μg/mL streptomycin (Life Technologies, Carlsbad, CA, USA), and 25 μg/mL amphotericin B (Life Technologies, CA, USA). All cultures were incubated at 37 °C in an atmosphere containing 5% CO_2_ and 95% air. Each group of 20 isolated hHFs was cultured in a medium containing adenosine at concentrations of 0.075, 0.25, and 0.45%. On days 3 and 6, the medium was replaced, and hHFs were photo-documented. The hair cycle stage of cultured human hair follicles was determined on days 0 and 6, according to hair cycle guidelines [58,59].

### 4.6. Immunohistochemistry of the Hair Follicle Organ

hHFs with treatment were fixed in 4% paraformaldehyde at room temperature for 10 min. hHF organ slices (3 μm thickness) were then permeabilized with phosphate-buffered saline containing 0.1% Triton x-100 and blocked with phosphate-buffered saline containing 5% FBS and 1% bovine serum albumin. After incubation with KI76 primary antibodies (1:200 dilutions, Abcam, Cambridge, UK) at 4 °C for 12 h and alkaline phosphatase-conjugated secondary antibodies (1:1000 dilution, Thermo Fisher Scientific, MA, USA) at room temperature for 1 h, BCIP/NBT (Sigma Aldrich, MO, USA) was used as substrate of alkaline phosphate. High-resolution images were taken using the EVOS™ FL Auto2 Imaging System (Thermo Fisher Scientific, Waltham, MA, USA).

### 4.7. Participants 

This study adhered to the tenets of the Declaration of Helsinki. The clinical studies were reviewed and approved by the Review Board of the LG Household and Healthcare Ltd. (approval numbers LGHH-20210520-AB-04-01 and LGHH-20211202-AB-04-01). Written informed consent was obtained from all participants before any study-related procedures. In total, 271 Korean volunteers participated in this study (Appendix A). Scalp parameters (hair density, hair thickness, scalp skin TEWL, and elasticity) were evaluated. Among the participants, 46 volunteers with male-pattern hair loss (n = 27) and female-pattern hair loss (n = 19) participated in the topical administration study (Appendix A). All volunteers were randomized into two groups: one group (n = 26) used a 5% MNX solution (Taiguk Pharm. Co., Ltd., Chungnam, Korea), while the other group (n = 20) received an adenosine complex (0.75% adenosine, 1% panthenol, and 2% niacinamide, APN) using a computer-generated random number system to maintain allocation concealment. The investigators and participants were blinded to the group allocation. The participants were supplied monthly with identical bottles containing similar textures, colors, and smells to maintain blinding. The agents (MNX and APN) were applied with daily topical administration (1 mL/day) to the frontal and vertex regions of the scalp. The total investigation period was four months. Details of randomization are provided in Appendix A. 

### 4.8. Evaluation of Scalp Parameters 

Baldness was evaluated by comparison with reference pictures according to the method described by dermatologists [60]. Hair density was measured using the phototrichogram technique with ×60 magnification [7,61]. Hair thickness was measured using image evaluation. The hair shafts were photo-imaged by ASW (Aram Hubis, Sungnam-si, Republic of Korea) at 200× magnification, and the shaft images (average 84.9 shafts in 1 cm^2^ area) were gathered at the same point in the parietal area of the scalp with semi-permanent marking. Images were evaluated using the ImageJ software 1.53 (NIH, Boston, MA, USA). TEWL was recorded in triplicate using a vapometer (Delfin Technologies, Kuopio, Finland) [62]. Measurements were performed after blow-drying and resting for 30 min in an air-controlled room. Skin elasticity was measured using a Ballistometer BLS 780 (Diastron Ltd., Hampshire, UK) to measure the material properties by interacting an impacting mass with the skin surface [63]. The five values obtained via automatic calculation are indentation, K, alpha (α), coefficient of restitution (CoR), and area. These parameters are correlated with both the hardness and elasticity of the skin [64]. All measurements were performed by the same investigator, who was blinded.

This study followed STROBE recommendations for reporting randomized clinical trials.

### 4.9. Statistical Analysis 

Statistical analyses were performed using Prism 10.0.2 (GraphPad Software, Boston, MA, USA). Data are presented as average ± standard deviation. We assessed the data for normal distribution and similar variance between groups. Statistical significance (* *p* < 0.05, ** *p* < 0.01, and *** *p* < 0.001) was evaluated using a two-tailed unpaired Student’s *t*-test for comparisons between two groups and a one-way analysis of variance (ANOVA) with relevant post hoc tests for multiple comparisons. Correlation analysis was performed using Pearson’s correlation test. All in vitro experimental data are presented as the average ± standard deviation of at least three independent experiments.

## 5. Conclusions

Hair growth is one of the highly energy-consuming processes in humans, with high energy expenditure, especially during the anagen stage [15]. Activating and stimulating energy metabolism through vitamin support will be beneficial for maintaining hair growth during the anagen phase. In this context, reinforcement of scalp and hair follicles with vitamins, such as panthenol, niacinamide, and biotin, can help the effects of MNX and adenosine to improve hair follicles and/or prevent alopecia.

In conclusion, our data demonstrated the anti-androgenic activity of adenosine. In addition, as an in vivo investigation, hair thickness distribution following alopecia severity and scalp physiology following topical administration of MNX and APN were examined. Adenosine showed anti-androgenic activity in vitro and ex vivo in the elongated anagen stages. Furthermore, both MNX and APN enhanced hair density and thickness, whereas adenosine decreased the vellus/terminal hair ratio. The adenosine complex enhanced scalp health and was more susceptible to hair thickness growth via anti-androgenic activity. We suggest that the evaluation of hair thickness distribution could help understand hair physiology and investigate new approaches for drug development.

## Figures and Tables

**Figure 1 ijms-25-06534-f001:**
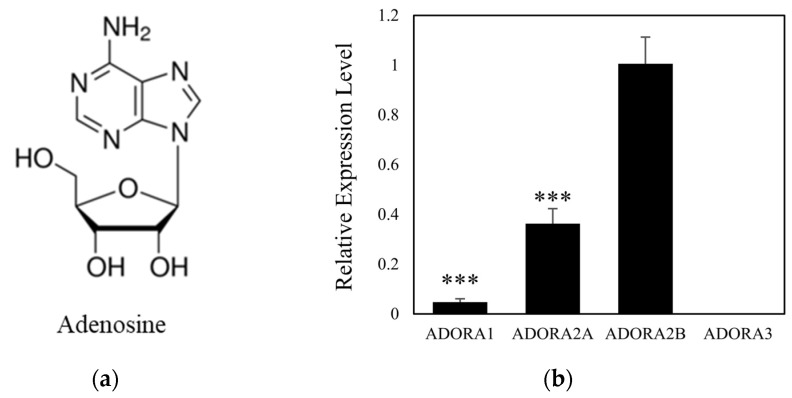
Chemical structures of adenosine and the mRNA expression profile of adenosine receptor subtypes. (**a**) Chemical structures. (**b**) The expression of four adenosine receptor subtypes, A_1_ (ADORA1), A_2A_ (ADORA2A), A_2B_ (ADORA2B), and A_3_ (ADORA3), was examined using quantitative RT-PCR analysis in cultured human dermal papilla cells. Ct values for each receptor subtype were indicated. ADORA3 was not detected (N.D) until 50 cycles. Significantly different compared with ADORA2B (*** *p* < 0.001). The experimental data were investigated in nine independent experiments.

**Figure 2 ijms-25-06534-f002:**
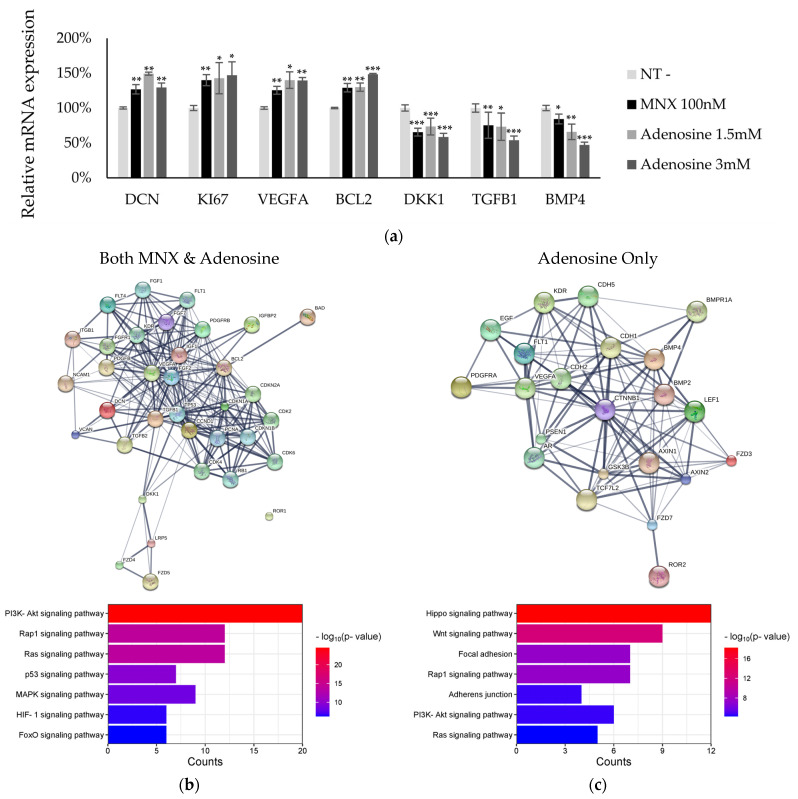
Signaling Pathway Activation by MNX and Adenosine treatment in cultured DPCs. The changes in mRNA expression profiles by MNX and adenosine in cultured human dermal papilla cells. Cells were treated with MNX and adenosine for one day, and mRNA expression was examined using RT-PCR. (**a**) Among 78 genes, the 7 genes with more than a 1.5-fold increase and statistical significance (*p* < 0.05) were displayed, red and blue colors mean increase and decrease in the mRNA expression, respectively. We further investigated using protein cluster analysis; (**b**) activated genes by both MNX and adenosine; and (**c**) activated genes by adenosine, not MNX. Significantly different compared with non-treated control (NT) (* *p* < 0.05, ** *p* < 0.01, *** *p* < 0.001). The experimental data were investigated in five independent experiments.

**Figure 3 ijms-25-06534-f003:**
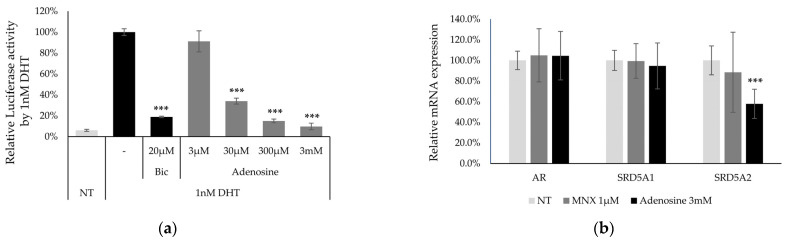
Anti-androgenic activity of adenosine. (**a**) The anti-androgenic activity of adenosine was further investigated using the 22Rv1-F5-MMTV/Luc cell line. (**b**) mRNA expression level of AR signaling (AR, SRD5A1, and SRD5A2) with adenosine treatment. (**c**,**d**) Phosphorylation of p53, Hsp27, JNK, MKK3, and MKK6 was evaluated. Significantly different compared with 1 nM DHT (*** *p* < 0.001) and non-treated (NT) control (# *p* < 0.05, ## *p* < 0.01, ### *p* < 0.001). The experimental data were investigated in at least six independent experiments.

**Figure 4 ijms-25-06534-f004:**
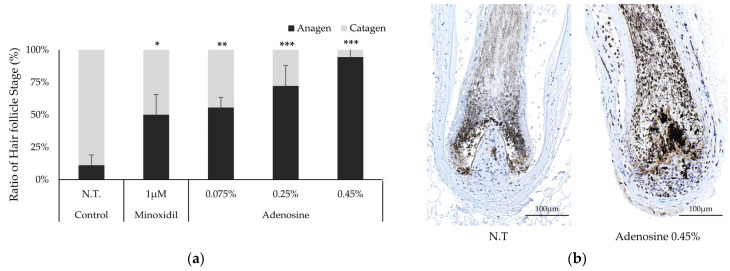
Effect of adenosine on anagen elongation in a human hair follicle organ culture. To evaluate the effect of adenosine, the anagen human hair follicle (hHF) was prepared and cultured for six days. Adenosine was treated at concentrations of 0.075%, 0.25%, and 0.45%. (**a**) At day 6, the cultured hair follicles were photo-documented. The percentage of hHF in an anagen or catagen state was determined. (**b**) Using immunohistochemistry, KI67 expression levels (dark blue dots) on hHF with adenosine treatment were evaluated. Melanin was visualized with black–brown dots. The hHFs with anagen morphology were examined. MNX was used as a positive control. The data represent the means of sixteen follicles. Significantly different compared with N.T (* *p* < 0.05, ** *p* < 0.01, *** *p* < 0.001). N.T, non-treated control. The experimental data were investigated in three independent experiments.

**Figure 5 ijms-25-06534-f005:**
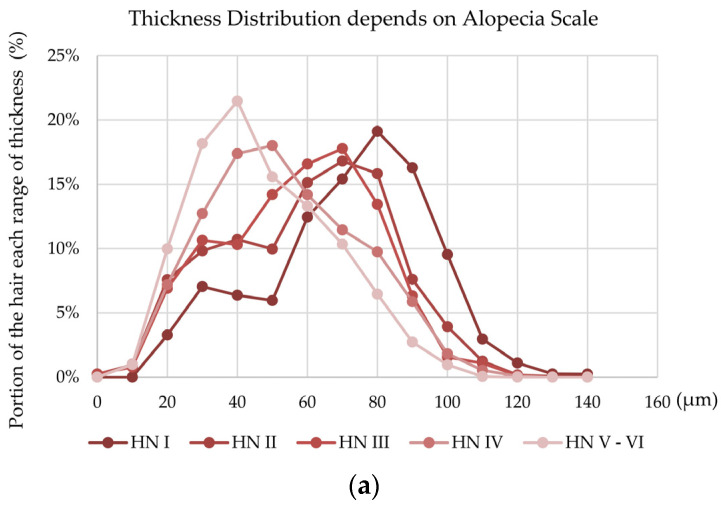
Hair thickness evaluation by alopecia severity and senescence. Hair thickness was evaluated, and the distribution pattern of hair shaft thickness was analyzed. (**a**) The hair thickness distribution of hair was displayed following alopecia severity. The scalp parameters of hair (**b**) density and (**c**) thickness were analyzed by alopecia severity. (**d**) The hair density and (**e**) thickness experience senescence and decrease with aging. Significantly different compared with HN-1 group (* *p* < 0.05, ** *p* < 0.01, *** *p* < 0.001) and 20s group (# *p* < 0.05, ### *p* < 0.001).

**Figure 6 ijms-25-06534-f006:**
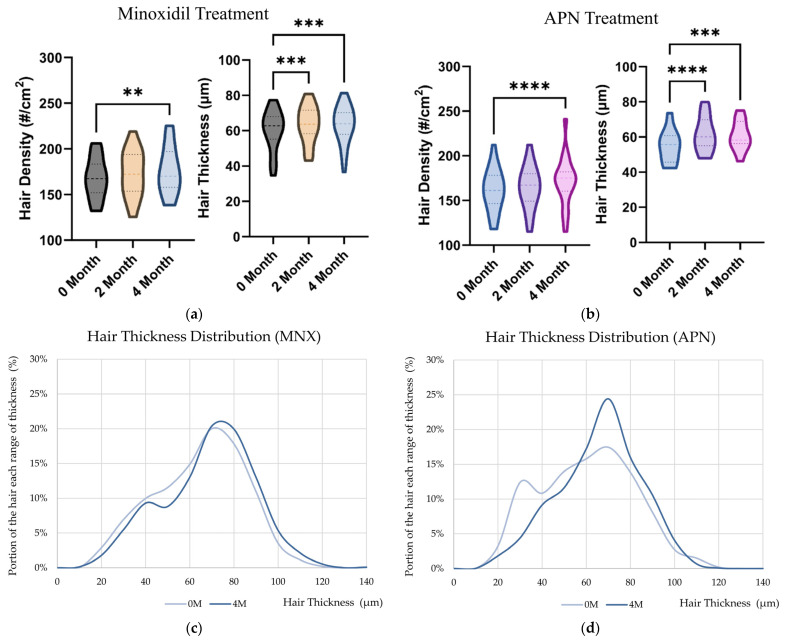
The hair thickness distribution changes with MNX and APN treatment. The change in hair density and thickness with the administration of (**a**) MNX and (**b**) APN were evaluated. The hair shaft thickness distribution changes were evaluated during the 4-month treatment with (**c**) MNX and (**d**) APN. (**e**) The representative crown images before and after the 4-month treatment of APN were displayed. Significantly different compared with the 0-month baseline (** *p* < 0.01, *** *p* < 0.001, **** *p* < 0.0001). #, number of hair shaft.

**Table 1 ijms-25-06534-t001:** Analysis of correlation genes in silico with activated genes in vitro by adenosine.

Node1	Node2	Combined Score
AR	CTNNB1	0.999
	GSK3B	0.987
	CDH1	0.913
CDH1	BMP4	0.903
CDH2	AXIN1	0.996
	CDH1	0.908
CDH5	KDR	0.999
	CDH2	0.921
	CDH1	0.919
PSEN1	CDH1	0.999
	GSK3B	0.996
	CDH2	0.903
TCF7L2	CTNNB1	0.999
	AXIN2	0.991
	LEF1	0.956
	AR	0.912

**Table 2 ijms-25-06534-t002:** Change in scalp parameters by senescence.

Scalp Parameters	Ages			
	20s	30s	40s	50~60s
Alopecia Scale (HN Scale)	1.73 (±1.05) ^(a)^	2.20 (±1.30) ^(b) ns^	3.70 (±1.11) ^(c)^ ***	3.97 (±0.72) ^(d)^ ***
Hair Density (number/cm^2^)	189.4 (±27.0) ^(a)^	178.9 (±28.1) ^(b)^ *	173.7 (±27.4) ^(c)^ *	172.0 (±31.6) ^(d)^ *
Hair Thickness (μm)	63.0 (±5.4) ^(e)^	59.2 (±13.2) ^(f)^ ***	53.1 (±13.1) ^(c)^ ***	60.8 (±11.8) ^(d)^ ***
Scalp Skin Barrier (TEWL) (g/m^2^h)	33.5 (±4.4) ^(g)^	34.6 (±5.8) ^(h)^ *	34.8 (±6.2) ^(i)^ *	
Scalp Elasticity (Area)	21.2 (±4.1) ^(j)^	22.2 (±3.4) ^(k) ns^	24.5 (±2.9) ^(l)^ *	

(a) Cohort in their 20s (n = 29); (b) cohort in their 30s (n = 89); (c) cohort in their 40s (n = 47); (d) cohort in their 50s–60s (n = 33); (e) cohort in their 20s (n = 26); (f) cohort in their 30s (n = 63); (g) cohort in their 20s (n = 15); (h) cohort in their 30s–40s (n = 37); (i) cohort in their 50s–60s (n = 21); (j) cohort in their 20s (n = 12); (k) cohort in their 30s to 40s (n = 11); and (l) cohort in their 50s–60s (n = 10). Significantly different compared with results of 20s (ns non-significant, * *p* < 0.05, *** *p* < 0.001).

**Table 3 ijms-25-06534-t003:** The change in hair density and thickness with administration of MNX and APN.

	Hair Density [Number/cm^2^]			Hair Thickness [μm]		
	0 Month	2 Month	4 Month	0 Month	2 Month	4 Month
MNX	168.1 ± 21.8	172.6 ± 24.0	176.5 ± 25.9 **	60.2 ± 11.1	63.4 ± 10.4 ***	63.3 ± 10.6 ***
APN	161.3 ± 22.7	164.2 ± 23.1	171.3 ± 27.0 ***	54.9 ± 8.9	61.3 ± 9.6 ***	60.5 ± 7.8 ***

Significantly different compared with 0 Month baseline (** *p* < 0.01, *** *p* < 0.001).

## Data Availability

The raw data supporting the conclusions of this article will be made available by the corresponding author upon request.

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
