# Peer review of "Hair Thickness Growth Effect of Adenosine Complex in Male-/Female-Patterned Hair Loss via Inhibition of Androgen Receptor Signaling"

_ijms, 2024, doi:10.3390/ijms25126534_

Round 1
Reviewer 1 Report
Comments and Suggestions for Authors
Author Response
Thank you for the advices and comments to the research.
Q1) In the introduction line 23-25 For 4 months administration, both MNX and APN group showed significant increase 23 in hair density (MNX + 5.01%, APN + 6.20%) and thickness (MNX + 5.14%, APN + 10.32%) can you mentioned p-vale for significance?
A1) We revised and add p-value on the abstract. The four values were statistically significant
Q2) Figure 1. Chemical structures of adenosine and the mRNA expression profile of adenosine receptor subtypes. (b) The expression of four adenosine receptor subtypes, A1 (ADORA1), A2A (ADORA2A), A2B (ADORA2B) and A3 (ADORA3) was examined using quantitative RT-PCR analysis in cultured human dermal papilla cell. Ct values for each receptor subtype were indicated. Statistical Significance missing?
A2) We revised and add p-value on the figure 1. The values were statistically significant
Q3) Table 2. Change of scalp parameters by senescence. Statistical Significance missing?
A3) We revised and add p-value on the figure 5 and table 2. The values were statistically significant.
Q4) Check typographic errors carefully.
A4) Thank you for the comments. We revised the manuscript.

Reviewer 2 Report
Comments and Suggestions for Authors
Dear authors,
This is an excellent paper about alopecia and the new possibility to treat hair loss. The introduction was well documented, the results and the conclusion are clearly presented, the quality of the images are good. I suggest to insert in the material and methods section the number of patients treated with MNX and with APN and the quantity of MNX and APN that was applied and the frequency/day.
Author Response
Thank you for the advice and comment to the research.
Q1) I suggest to insert in the material and methods section the number of patients treated with MNX and with APN and the quantity of MNX and APN that was applied and the frequency/day.
A1) Thank you for the response.
I revised the number of patients and quantity of investigation in material method section.

Reviewer 3 Report
Comments and Suggestions for Authors
My detailed comments are as following:
1. Figure 1b: The authors should indicate the number of repeats performed for this experiment. They should include the statistical analysis results in Figure 1b. Additionally, a housekeeping gene such as GAPDH should be analyzed as a control. The CT values for ADORA1/2A/2B/3 are over 30, which raises concerns about whether these high values are due to the chosen primers or the RNA quality.
2. Figure 2: The authors need to explain why they focused on these 78 genes and the criteria for selecting them. All gene names should be clarified in the supplemental table. In the main text, the authors should list the genes that showed significant changes upon treatment with minoxidil or adenosine.
3. Figure 2b/c: Consider using figures instead of tables to show the enriched pathways. These figures should reflect the number of genes in each pathway, p-values, and expression changes (increases or decreases). The authors can refer to the following link for examples of how these figures would look like:
https://yulab-smu.top/biomedical-knowledge-mining-book/enrichplot.html
4. Figure Legends: The figure legends should be detailed, including the number of biological repeats performed.
5. Figure 3b: In addition to examining SRD5A1/5A2, the authors should also investigate whether the genes responding to adenosine treatment in Figure 2a are affected by the anti-androgen activity of adenosine.
6. Figure 6a/b: While the effects of minoxidil or APN treatment on hair density and hair thickness are statistically significant, they appear to be minor. The authors should discuss the biological significance of this statistical significance.
Comments on the Quality of English LanguageMinor revisions should be performed before re-submission.
Author Response
Thank you for the advices and comments to the research.
Q1). Figure 1b: The authors should indicate the number of repeats performed for this experiment. They should include the statistical analysis results in Figure 1b. Additionally, a housekeeping gene such as GAPDH should be analyzed as a control. The CT values for ADORA1/2A/2B/3 are over 30, which raises concerns about whether these high values are due to the chosen primers or the RNA quality.
A1) We re-evaluated the samples with higher cDNA amount and revised the results.
To reduce concerns about Ct values (over 30), we re-examined the RT-PCR with 2μg cDNA samples and updated the figure results. GAPDH also attached as internal control. The evaluation was repeated with nine independent experiments, and statistical analysis also performed. The results were revised in figure 1.
Q2). Figure 2: The authors need to explain why they focused on these 78 genes and the criteria for selecting them. All gene names should be clarified in the supplemental table. In the main text, the authors should list the genes that showed significant changes upon treatment with minoxidil or adenosine
A2) We examined 78 genes with 6 categories; Growth factor signaling, Tissue development, Cell differentiation, Cell junction, Cytoskeleton, and Cellular survival. Detail lists of 78 gene name were added with descriptions in Supplementary Figure S1.
Q3). Figure 2b/c: Consider using figures instead of tables to show the enriched pathways. These figures should reflect the number of genes in each pathway, p-values, and expression changes (increases or decreases). The authors can refer to the following link for examples of how these figures would look like:
https://yulab-smu.top/biomedical-knowledge-mining-book/enrichplot.html
A3) Following the comment, we revised the figure 2 b,c.
Q4). Figure Legends: The figure legends should be detailed, including the number of biological repeats performed.
A4) We described the number of biological repeats in figure legends.
Q5). Figure 3b: In addition to examining SRD5A1/5A2, the authors should also investigate whether the genes responding to adenosine treatment in Figure 2a are affected by the anti-androgen activity of adenosine.
A5) To understand effect of adenosine to androgen receptor signaling, we furtherly investigated about kinase which correlated with androgen receptor stability and activity.
Furthermore, phosphorylation of kinase proteins (p53, Hsp27, JNK, and MKK3/6), which were contributed to androgen receptor pathway (1), were evaluated (Figure 3c). As the elements which is negatively correlated with androgen receptor pathway, p53 (2), MKK3/6 (1, 3) was examined. In addition, p-JNK (4, 5) and Hsp27 (6) was examined as the elements which are positively correlated with androgen receptor. Even if the reports were based on prostate cancer, the phosphorylation increase of p53, MKK3/6, and phosphorylation decrease of Hsp27 and JNK could be correlated with down-regulation of androgen receptor pathway. In addition, DKK1, reported to upregulated by androgen receptor (7), were decreased by adenosine. On the contrary, Decorin, which inhibited by AR activity (8), significantly increased by adenosine.
We also understand that this results can not fully elucidate anti-androgenic activity of adenosine in human hair follicle. As further research, it could be meaningful investigation about androgen receptor pathway with siRNA and inhibitor study in hDCPs.
Ref 1) Shah, K.; Bradbury, N. A., Kinase modulation of androgen receptor signaling: implications for prostate cancer. Cancer cell & microenvironment 2015, 2, (4).
Ref 2) Alimirah, F.; Panchanathan, R.; Chen, J.; Zhang, X.; Ho, S.-M.; Choubey, D., Expression of androgen receptor is negatively regulated by p53. Neoplasia 2007, 9, (12), 1152-1159.
Ref 3) Gioeli, D.; Black, B. E.; Gordon, V.; Spencer, A.; Kesler, C. T.; Eblen, S. T.; Paschal, B. M.; Weber, M. J., Stress kinase signaling regulates androgen receptor phosphorylation, transcription, and localization. Molecular endocrinology (Baltimore, Md.) 2006, 20, (3), 503-15.
Ref 4) Huang, P.-H.; Wang, D.; Chuang, H.-C.; Wei, S.; Kulp, S. K.; Chen, C.-S., α-Tocopheryl succinate and derivatives mediate the transcriptional repression of androgen receptor in prostate cancer cells by targeting the PP2A-JNK-Sp1-signaling axis. Carcinogenesis 2009, 30, (7), 1125-1131.
Ref 5) Xu, R.; Hu, J., The role of JNK in prostate cancer progression and therapeutic strategies. Biomedicine & Pharmacotherapy 2020, 121, 109679.
Ref 6) Zoubeidi, A.; Zardan, A.; Beraldi, E.; Fazli, L.; Sowery, R.; Rennie, P.; Nelson, C.; Gleave, M., Cooperative interactions between androgen receptor (AR) and heat-shock protein 27 facilitate AR transcriptional activity. Cancer Res 2007, 67, (21), 10455-65.
Ref 7) Choi, Y. H.; Shin, J. Y.; Kim, J.; Kang, N. G.; Lee, S., Niacinamide Down-Regulates the Expression of DKK-1 and Protects Cells from Oxidative Stress in Cultured Human Dermal Papilla Cells. Clin Cosmet Investig Dermatol 2021, 14, 1519-1528
Ref 8) Montano, M.; Dinnon, K. H.; Jacobs, L.; Xiang, W.; Iozzo, R. V.; Bushman, W., Dual regulation of decorin by androgen and Hedgehog signaling during prostate morphogenesis. Developmental Dynamics 2018, 247, (5), 679-685.
Q6) Figure 6a/b: While the effects of minoxidil or APN treatment on hair density and hair thickness are statistically significant, they appear to be minor. The authors should discuss the biological significance of this statistical significance.
A6) We described the in-vivo results comparing with other reports about topical minoxidil in discussion.
In previous reports (Ramos et al., 2020), 5% topical minoxidil promoted to increase total hair density from 163.2 (±46.0) to 176.3 (±61.5) number/cm2, but density increase of terminal hair was limited (from 113.3 (±41.1) to 116.8 (±44.9) number/cm2) for six-month administration.
Following table 3, hair density is increased in both MNX (from 168.1 to 176.5 number/cm2) and APN group (from 168.1 to 176.5 number/cm2). In addition, in MNX group of figure 6, the ratio between vellus hair and terminal hair did not change significantly.
Even if it could be difficult to direct comparison between alopecia cohort from Asia and Caucasian, total hair density increase of minoxidil between this study and previous reports can be reasonable. (Table 3 MNX group: 168.1 → 176.5 / Ramos et al., 2020 Topical MNX: 163.2 → 176.3 number/cm2)
However, as following the reviewer’s comment, the recovery of alopecia by both minoxidil and adenosine complex treatment are too slow and minor compared with severity of alopecia. On the contrary, hair thickness was significantly increased by 2-month administration (Table 3 MNX group: 60.2 → 63.4μm / APN group: 54.9 → 61.3μm). There is possibility that mode of mechanism could be different between hair density and thickness.
In this study, investigation period (4 month) was limited to compare with other reports (6month, 1 year, and sometimes 4year). The further research about long-term investigation (1 year or more) about minoxidil and adenosine complex could help to understand scalp physiology about hair thickness growth.
We added and revised discussion part about comparison with other clinical reports.
Ref) Ramos, Paulo Müller, et al. "Minoxidil 1 mg oral versus minoxidil 5% topical solution for the treatment of female-pattern hair loss: a randomized clinical trial." Journal of the American Academy of Dermatology 82.1 (2020): 252-253.

Reviewer 4 Report
Comments and Suggestions for Authors
In the study entitled “Hair thickness growth effect of adenosine complex in male-/female-patterned hair loss via inhibition of androgen receptor signaling.” the authors report on the comparison of usage of minoxidil 5% and a formulation based on 0.75% Adenosine, 1% Panthenol and 2% Niacinamide in patients with alopecia for four months. Some aspects of mode of action of adenosine yet investigated by the authors in other studies were further clarified in vitro.
The study is interesting and well presented. I have some remarks and suggestions for the authors:
- In the last paragraph of the introduction, the authors need to clearly state the novelty and importance of this paper together with detailed aims and prospects of this study.
- There are some typos mistakes throughout the manuscript. The author needs to revise these carefully.
- All the data presented in paragraphs 2.2 and 2.3 should be in depth validated using other techniques (western blot, immunoprecipitation, interactome analysis by mass spectrometry, cellular cycle assay, apoptosis assays and so on…). Report and discuss these pathways only on the basis of transcriptomic analysis and in silico prediction is not sufficient. Where is possible support your data with literature data.
- Comparison with other similar in vitro and especially clinical study
- Add a conclusion section starting sentences from line 302-311. It is a good starting.
Materials and methods
- Add (Company, City, Country) for all instrumentations reagents standards, software. Standardize and change it in all manuscript.
- Please add some additional details about the usage tips (e.g. daily application …) provided to the participants.
Based on these comments I strongly encourage the authors to improve the manuscript, since it is a suitable candidate for publication in International Journal of Molecular Sciences.
Comments on the Quality of English LanguagePlease see the report.
Author Response
Thank you for the advices and comments to the research.
Q1) In the last paragraph of the introduction, the authors need to clearly state the novelty and importance of this paper together with detailed aims and prospects of this study.
A2) Following the comment, we revised the last paragraph of the introduction.
Q2) There are some typos mistakes throughout the manuscript. The author needs to revise these carefully.
A2) Following the comment, we revised manuscript.
Q3) All the data presented in paragraphs 2.2 and 2.3 should be in depth validated using other techniques (western blot, immunoprecipitation, interactome analysis by mass spectrometry, cellular cycle assay, apoptosis assays and so on…). Report and discuss these pathways only on the basis of transcriptomic analysis and in silico prediction is not sufficient. Where is possible support your data with literature data.
A3) To understand effect of adenosine to androgen receptor signaling, we furtherly investigated about kinase which correlated with androgen receptor stability and activity.
Furthermore, phosphorylation of kinase proteins (p53, Hsp27, JNK, and MKK3/6), which were contributed to androgen receptor pathway (1), were evaluated (Figure 3c). As the elements which is negatively correlated with androgen receptor pathway, p53 (2), MKK3/6 (1, 3) was examined. In addition, p-JNK (4, 5) and Hsp27 (6) was examined as the elements which are positively correlated with androgen receptor. Even if the reports were based on prostate cancer, the phosphorylation increase of p53, MKK3/6, and phosphorylation decrease of Hsp27 and JNK could be correlated with down-regulation of androgen receptor pathway.
We also understand that this results can not fully elucidate anti-androgenic activity of adenosine in human hair follicle. As further research, it could be meaningful investigation about androgen receptor pathway with siRNA and inhibitor study in hDCPs.
Ref 1) Shah, K.; Bradbury, N. A., Kinase modulation of androgen receptor signaling: implications for prostate cancer. Cancer cell & microenvironment 2015, 2, (4).
Ref 2) Alimirah, F.; Panchanathan, R.; Chen, J.; Zhang, X.; Ho, S.-M.; Choubey, D., Expression of androgen receptor is negatively regulated by p53. Neoplasia 2007, 9, (12), 1152-1159.
Ref 3) Gioeli, D.; Black, B. E.; Gordon, V.; Spencer, A.; Kesler, C. T.; Eblen, S. T.; Paschal, B. M.; Weber, M. J., Stress kinase signaling regulates androgen receptor phosphorylation, transcription, and localization. Molecular endocrinology (Baltimore, Md.) 2006, 20, (3), 503-15.
Ref 4) Huang, P.-H.; Wang, D.; Chuang, H.-C.; Wei, S.; Kulp, S. K.; Chen, C.-S., α-Tocopheryl succinate and derivatives mediate the transcriptional repression of androgen receptor in prostate cancer cells by targeting the PP2A-JNK-Sp1-signaling axis. Carcinogenesis 2009, 30, (7), 1125-1131.
Ref 5) Xu, R.; Hu, J., The role of JNK in prostate cancer progression and therapeutic strategies. Biomedicine & Pharmacotherapy 2020, 121, 109679.
Ref 6) Zoubeidi, A.; Zardan, A.; Beraldi, E.; Fazli, L.; Sowery, R.; Rennie, P.; Nelson, C.; Gleave, M., Cooperative interactions between androgen receptor (AR) and heat-shock protein 27 facilitate AR transcriptional activity. Cancer Res 2007, 67, (21), 10455-65.
Q4) Comparison with other similar ssin vitro and especially clinical study
A4) We described the in-vivo results comparing with other reports about topical minoxidil in discussion.
In previous reports (Ramos et al., 2020), 5% topical minoxidil promoted to increase total hair density from 163.2 (±46.0) to 176.3 (±61.5) number/cm2, but density increase of terminal hair was limited (from 113.3 (±41.1) to 116.8 (±44.9) number/cm2) for six-month administration.
We add discussion part about comparison with other clinical reports
Ref) Ramos, Paulo Müller, et al. "Minoxidil 1 mg oral versus minoxidil 5% topical solution for the treatment of female-pattern hair loss: a randomized clinical trial." Journal of the American Academy of Dermatology 82.1 (2020): 252-253.
Q5) Add a conclusion section starting sentences from line 302-311. It is a good starting.
A5) Following the comment, we added conclusion section after discussion section.
Materials and methods
Q6) Add (Company, City, Country) for all instrumentations reagents standards, software. Standardize and change it in all manuscript.
A6) Following the comment, we revised materials and methods section.
Q7) Please add some additional details about the usage tips (e.g. daily application …) provided to the participants.
A7) Following the comment, we updated materials and methods section.
The minoxidil and adenosine complex were performed following the guidance;
Usage - One a day, 1ml/day; Period: 4 –month; Site: Frontal and Vertex region of scalp

Round 2
Reviewer 4 Report
Comments and Suggestions for Authors
The authors addressed the most part of the reviewer's comments and suggestions. I still remain of idea that the manuscript could be better with further experiments elucudating mechanism of adenosine. However I will suggest only minor revisions. Some improvements in the English language are necessary.
Comments on the Quality of English LanguageThe authors addressed the most part of the reviewer's comments and suggestions. I still remain of idea that the manuscript could be better with further experiments elucudating mechanism of adenosine. However I will suggest only minor revisions. Some improvements in the English language are necessary.
Author Response
Thank you for the comments.
We revised manuscript with English editing Services.